# Comparative Study on Thermal Response Mechanism of Two Binders during Slow Cook-Off

**DOI:** 10.3390/polym14173699

**Published:** 2022-09-05

**Authors:** Xinzhou Wu, Jun Li, Hui Ren, Qingjie Jiao

**Affiliations:** 1State Key Laboratory of Explosion of Science and Technology, Beijing Institute of Technology, Beijing 100081, China; 2Science and Technology on Aerospace Chemical Power Laboratory, China Aerospace Science and Technology Corporation, Xiangyang 441003, China

**Keywords:** HTPB, HTPE, slow cook-off, mechanical properties, microstructure

## Abstract

The HTPE (hydroxyl-terminated polyether) propellant had a lower ignition temperature (150 °C vs. 240 °C) than the HTPB (hydroxy-terminated polybutadiene) propellant in the slow cook-off test. The reactions of the two propellants were combustion and explosion, respectively. A series of experiments including the changes of colors and the intensity of infrared characteristic peaks were designed to characterize the differences in the thermal response mechanisms of the HTPB and HTPE binder systems. As a solid phase filler to accidental ignition, the weight loss and microscopic morphology of AP (30~230 °C) were observed by TG and SEM. The defects of the propellant caused by the cook-off were quantitatively analyzed by the box counting method. Above 120 °C, the HTPE propellant began to melt and disperse in the holes, filling the cracks, which generated during the decomposition of AP at a low temperature. Melting products were called the “high-temperature self-repair body”. A series of analyses proved that the different thermal responses of the two binders were the main cause of the slow cook-off results, which were likewise verified in the propellant mechanical properties and gel fraction test. From the microscopic point of view, the mechanism of HTPE’s slow cook-off performance superior to HTPB was revealed in this article.

## 1. Introduction

Solid propellant, the fuel of a solid engine, is playing an increasingly important role in solid propulsion technology and modern aerospace technology because of its simple structure, convenient storage, short launch preparation time and strong mobility. In the international community, the chemical names of binders commonly represent solid propellants. Binder is mainly used to uniformly mix oxidants, plasticizers, curing agents and so on, so that the slurry of propellant can obtain certain rheological properties [1]. With the developments of high energy and low vulnerability of propellants, the binder has become an important factor in safety regulation.

Hydroxy-terminated polybutadiene (HTPB) is a liquid prepolymer [2], and due to its excellent physical properties, such as low viscosity, low glass transition temperature [3], high mechanical and thermal stability, lifting the load of solid propellant, etc. [4], it is widely used as a binder to make solid propellants have a stable burning rate and low-pressure exponent. HTPB bonds aluminum powder and ammonium perchlorate (AP) solid powder [5,6], under the action of isocyanate (TDI), propellant crosslinking, curing and forming a certain geometric shape and mechanical strength [7,8]. Compared with the carboxyl-terminated polybutadiene (CTPB) binder, HTPB shows better extensibility at a low temperature, and exhibits better thermodynamic and anti-aging properties [9]. HTPB does not contain any energetic groups; it has no contribution to improving the energy output and combustion rate of the solid propellant. So, the research of HTPB focuses on how to enhance energy with modifications. Under the condition of keeping the mechanical properties of HTPB, experts try to introduce various high-energy groups into the polymer backbone, hoping to further improve the overall energy of the propellant [10,11,12,13,14,15]. Toward the functionalization goal, the preparation of a hydroxyl-terminated polystyrene-b-polybutadiene-b-polystyrene triblock copolymer (SBS) with high cis-1, 4 content via a novel nickel catalyst was reported. A hydroxyl group was successfully introduced at the end of the triblock copolymer (HO–SBS–OH) by FT-IR, ^1^H-NMR and ^13^C NMR methods [16].

As of recently, with the requirement of low vulnerability of the propellant, missile weapons should withstand six examinations, i.e., a fast cook-off test, slow cook-off test, gap test, bullet impact test, fragment impact test and jet test according to the U.S. MIL-STD-2105D and NATO STANAG 4439 [17]. The HTPB propellant failed to pass the slow cook-off test and exploded, which was the most rigorous low vulnerability test. Compared with the HTPB propellant, the HTPE (hydroxyl-terminated polyether) propellant composed of hydroxyl-terminated polyether only combusted in the slow cook-off test, and the response degree declined. Someone believed that the HTPE binder introduced a block structure in the molecular chain, which generated low molecular oligomers during thermal decomposition, and then delayed the formation of combustible molecules, thus reducing the response characteristics of propellants to the slow cook-off test [18,19]. HTPE has a tendency to replace HTPB in solid propellants so as to pass slow cook-off experiments.

Up to now, the mechanism of HTPB and HTPE binders in a slow cook-off test lacked the analysis detailed. Most analyses focused on these factors such as charge size [20,21], and concluded that AP is the main factor during the slow cook-off of HTPB and HTPE propellants [22,23]. However, little research studies the durability and degradation of polymeric materials during the slow cook-off process, especially the changes in the mechanical properties [24,25] and decomposition path [26] of binders. It is an interesting topic to study the change pattern of a small number of polymeric materials in a propellant slow cook-off, and to study the effect of the decomposition of polymeric materials on the overall slow cook-off result from a microscopic perspective. In this article, HTPB and HTPE propellants used in propellant charge were compared with a small slow cook-off test instrument. Igniting temperatures of the two solid propellants, the decomposing mechanism of HTPB and HTPE and the interaction reactions between a binder and AP (Ammonium perchlorate) were analyzed. From the micro-view, differences between the two binders during the slow cook-off response to thermal stimulation were investigated in order to explain how the HTPE propellant could improve the safety compared with the HTPB propellant.

## 2. Materials and Methods

### 2.1. Materials and Instruments

The HTPB binder(Ⅲ) was purchased from the Liming Chemica Research Institute of Chemical Industry(Luoyang, China). The HTPE binder was purchased from the Science and Technology on Aerospace Chemical Power Laboratory(Xiangyang, China). TDI (Industrial grade) was purchased from the School of Chemistry and Chemical Engineering Inner Mongolia University(Hohhot, China). Aluminum powder (FLQT1 grade) was purchased from Angang industrial micro aluminum powder Co., Ltd. (Anshan, China). AP(Ⅲ) was purchased from Dalian north potassium chlorate Co., Ltd. (Dalian, China). The molecular formulas of the two binders and curing agents are shown in Figure 1.

A scanning electron microscope (SEM), JSM-6360LV, Voltage 15 kV, working distance 12 mm, vacuum 1.0 × 10^−5^ Pa was obtained from Japan Co., Ltd. (Tokyo, Japan). An optical microscope, MEF4M, was obtained from Germany Leica Microsystems Ltd. (Wetzlar, Germany). A Fourier transform infrared spectrometer (FTIR), Model 5700, was obtained from the American Thermal Electron Corporation(Waltham, MA, USA). A cupping machine, INSTRON4202, was obtained from the American Instron Corporation (Boston, MA, USA). A small cook-off test instrument, HTSSC I-002, was made by the Science and Technology in Aerospace Chemical Power Laboratory (Xiangyang, China). The mass of the test sample is 1~1000 g, the temperature control accuracy is ±1 °C and the experimental temperature range is 20~350 °C. A thermogravimetric differential scanning calorimeter (TG-DSC), SDTQ600, was obtained from American TA Instruments (New Castle, DE, USA).

### 2.2. Methods

Samples preparation: the HTPB binder and TDI curing agent were mixed evenly in a beaker according to the curing parameter of 1.0 [27], poured into a special mold, defoamed under a vacuum, cured at a constant temperature of 50 °C for 168 h and demolded after curing to make the shape required for the experiment. Adjusting the curing parameter to 1.3, the HTPE binder film was prepared by the same process.

Propellant preparation: samples were prepared by using a vertical kneader, and the binder and oxidizer were mixed in a certain order and ratio (HTPB propellant formulation ratio: HTPB binder/AP/Al/auxiliary = 10/70/18/2, HTPE propellant formulation ratio: HTPE binder/AP/Al/auxiliary = 20/72/5/3) at a mixing temperature of 50~55 °C. These samples were mixed for 120 min and cast under a vacuum, where the casting temperature was 50~55 °C. Then, they were cured at a constant temperature of 50 °C for 168 h.

Slow cook-off test: the response characteristics of the HTPB and HTPE scaled-down solid engines with a specimen size of Ф100 mm × 200 mm were studied. The thickness of the shell and end cover was 3 mm, the material was 45# steel and the heating rate was 3.30 °C/h. Meanwhile, the HTPB and HTPE propellants were tested at a heating rate of 3.30 °C/h in a small cook-off test apparatus. The microstructure of propellants at different temperatures during the cook-off process was analyzed by a scanning electron microscope.

Material tensile test: the propellant samples were processed into standard dumbbell-shaped dimensions and analyzed for modulus, maximum elongation and maximum tensile strength at different slow cook-off temperatures. Considering the different ignition temperatures of different solid propellants [28], the HTPB propellant was tested below 220 °C and the HTPE propellant was tested below 140 °C. The samples were taken out and cooled to room temperature, and tensile tests were carried out at a speed of 100 mm/min. In the mechanical test, three groups of tests were completed and the group with larger deviation was removed.

Gel percentage test: the solid propellant was processed into 10 mm × 10 mm × 1 mm thin sheets, and about 3 g samples were weighed and placed in a flask at the grinding mouth. Then, 100 mL trichloromethane solution was added and the sample was soaked for about 12 h. The sample and extract were transferred to a Soxhlet extractor, and 100 mL trichloromethane solution was added into a single-mouth flask, and heated in a water bath at 80 °C for 6 h. The extracted samples were transferred to a surface dish, placed in a fume hood to remove a large amount of solvent and then transferred to a vacuum oven to maintain 40 °C for drying under vacuum. After drying for 1 h, the sample was taken out and placed in the dryer to cool to room temperature before weighing. The operation was repeated until the weight of the specimens remained the same.

Infrared spectrum test: a Fourier transform infrared spectrometer was used to scan 20 times in the air atmosphere for spectral accumulation. The scanning range was from 4000 to 400 cm^−1^ with a resolution of 4 cm^−1^; the heating range was from 20 to 320 °C with a heating rate of 2 °C/min. The functional groups of HTPB and HTPE propellants were tested at room temperature and slow cook-off progress, respectively, and the changes in these functional groups’ contents with temperature were analyzed.

Thermogravimetric analysis test: 2~5 mg samples were experimented on under the air atmosphere and different heating rate. The experimental accuracy was about 0.1 μg, and the experimental temperature range was from room temperature to 500 °C with a heating rate of 0.1 °C/min to 50 °C/min.

### 2.3. Slow Cook-Off Test on a Scaled-Down Solid Engine

Figure 2a,b show the physical and schematic diagrams, respectively. Temperature sensors were placed to record the temperature field and the temperature change of the solid propellant. They were placed on the outer surface and center of the Φ 100 mm × 200 mm scaled-down solid engine. To improve experimental efficiency, the HTPB propellant was heated to 70 °C at a heating efficiency of 1.0 °C/min firstly and then heated at a heating rate of 3.30 °C/h. To further shorten the experimental period, the HTPE reduced-ratio solid engine was heated to 100 °C at a rate of 2.0 °C/min firstly and then at 3.30 °C/h during the slow cook-off test.

It can be seen from Figure 3a that the temperature field of the reduced-ratio solid engine was very uniform in the slow heating process of 3.30 °C/h. By linear fitting, the heating rate of the HTPB scaled-down solid engine was 3.31 °C/h, and that of the HTPE scaled-down solid engine was 3.29 °C/h. The heating rates all met the experimental requirement (<0.5 °C/h). The internal temperature of the HTPB scaled-down solid engine increased sharply at 55 h and responded at 244 °C. The temperature of the HTPE scaled solid engine increased sharply at 13 h, and the ignition temperature was determined to be 150 °C by tangency. Figure 3b shows the field results of the HTPB scaled-down solid engine after the end of response. The shell barrel section was torn and fully expanded along the axial direction, and the end cap was broken from the threads. There was no residual charge at the site, and the shell showed signs of ablation. There were no explosive craters on the ground. Two of the witness plates had more than one obvious dent, but none of them were perforated and the remaining two plates had no obvious dents and perforations. Collectively, the response level of the HTPB propellant in the slow cook-off test was determined as an explosion. Figure 3c shows the field results after the response of the HTPE scaled-down solid engine, which shows that the shell barrel was intact, one side of the end cover was intact and the other side was torn in half. There was no residual charge at the scene, and the shell had traces of ablation. There were no craters on the ground and no dents and perforations in the four witness plates. The experimental results were judged to be combustion.

The experiments with the Ф100 mm × 200 mm reduced-ratio engine show that the ignition temperature of the HTPB propellant was about 244 °C and the response grade was an explosion. The ignition temperature of the HTPE propellant was about 150 °C, and the response level was lower than HTPB, which was combustion only.

## 3. Results and Discussion

The fuel in solid engines generally consists of Al, AP and binder systems, and the ignition temperature of a slow cook-off is between 100 and 300 °C. The oxidation temperature of Al is above 600 °C [29,30], which is much higher than the ignition temperature of a solid engine in a slow cook-off test, indicating that Al powder mainly acts as a heat conductor in a solid engine and has little influence on slow cook-off temperature. Therefore, the main factors affecting the ignition of the solid propellant were studied about AP and binder components, respectively.

### 3.1. Response Behavior of AP during Slow Heating

The content of AP in solid engine charge is the highest and more than 70% in some solid propellants [31]. Other studies have shown that the thermal decomposition mechanism of AP is mainly concentrated in two stages: the low-temperature stage (250–350 °C) and the high-temperature stage (>350 °C). The main mechanisms are the electron transfer mechanism [32,33] and the proton transfer mechanism [34,35]. In the study of the main factors of solid engine ignition, the low-temperature decomposition stage of AP was mainly concerned.

The TG-DSC curves and reaction activation energy of AP were analyzed. The DSC curve showed an endothermic peak and two exothermic peaks, among which, the endothermic peak was 243.2 °C, and this temperature range was the crystal transition peak (rhombic crystal to cubic crystal) of AP. The DSC curve showed that the ignition temperature of AP was 294.6 °C. Kinetic parameters of AP were calculated by the Ozawa method and the Kissinger method: they were 117.8 kJ/mol and 106.1 kJ/mol, respectively.

According to the characteristics of the slow cook-off test, this paper focused on the decomposition characteristics of AP at 30~250 °C. The weight loss rate of AP at different temperatures is shown in Figure 4. AP began to produce trace weight loss at 150 °C. With the increase in temperature, the weight loss rate increased gradually. When the temperature was higher than 200 °C, the weight loss rate increased rapidly. When the temperature reached 230 °C, the weight loss rate reached 31.5%.

The morphologies of AP at a low temperature were studied with SEM. The experimental results are shown in Figure 5. The surface of AP was smooth below 100 °C. When the temperature continued to rise to 190 °C, no visible defects and holes appeared on the surface. When the temperature continued to increase to 200 °C, some tiny holes appeared on the surface. Finally, when reaching 230 °C, holes and defects appeared obviously.

AP is the main gas-producing substance in the solid propellant, and its reaction affects the response grade of a solid propellant in a slow cook-off test directly. Through exploring the decomposition temperature of AP, it was found that AP will decompose violently only when it is above 200 °C, which is higher than the response temperature of the HTPE propellant and close to that of the HTPB propellant. It is speculated that AP is the main ignition factor of HTPB propellant.

### 3.2. Thermal Response of HTPB Film at Different Temperatures

In order to study the ignition characteristics of the HTPB film cured by TDI in the slow cook-off condition, DSC was used to study the ignition temperature of the HTPB film. The DSC curve of the HTPB film began to climb at about 320.5 °C, followed by a strong exothermic peak at 367.7 °C. The TG curve shows that the HTPB film began to lose weight at 282 °C and the weight loss rate reached 34% at 430 °C. The Ozawa method and Kissinger method were used to calculate the kinetic parameters of the HTPB film’s exothermic peak as 161.3 kJ/mol and 159.7 kJ/mol, respectively.

The surface of the HTPB film was observed at different temperatures in the process of the slow cook-off. The experimental results are shown in Figure 6. The HTPB film at room temperature (20 °C) was light yellow translucent. With the temperature increasing, the color of the HTPB film gradually deepened, and after 150 °C, the rate of the color deepening was obvious. When it reached 190 °C, the whole film had become brown, and all of it became dark brown at 200 °C. The reason for the deepening of the color may be that the colored groups were produced by the decomposition of the HTPB film, and they increased with the rising temperature.

The total reflection infrared test results of the HTPB film at different temperatures are shown in Figure 7. The stretching vibration peak of C–O was at 1076 cm^−1^, the vibration peak of C=C on the benzene ring was at 1533 cm^−1^ and the N–H vibration peak was at 1537 cm^−1^.

The changes of various groups in the HTPB film during the slow cook-off were analyzed, especially the changes in the N–H bond and C–O bond. The change of the N–H bond with temperature is shown in Figure 8a. From room temperature to 200 °C, the N–H bond was divided into two stages. The first stage was between room temperature and 120 °C, and the peak strength of N–H was maintained. When the temperature was higher than 120 °C, the strength of the N–H bond decreased gradually, and when the temperature was higher than 170 °C, the strength of the N–H bond decreased significantly. It is believed that the breakdown of the urethane bond in the HTPB film during the slow cook-off process resulted in the decrease in the peak strength of the N–H bond. The breaking of the N–H bond resulted in the formation of amino or alkyl radicals and the release of CO_2_. After the breaking of the N–H bond, it may rearrange and form an aniline structure. The reaction process is shown in Figure 8b.

The variation of the C–O bond at different temperatures in the process of the slow cook-off is shown in Figure 9a. The C–O bond was also divided into two stages. The first stage is that the peak strength of C–O was maintained between room temperature and 170 °C. In the second stage, when the temperature was higher than 190 °C, the strength of the C–O bond gradually weakened, and when the temperature was higher than 190 °C, the strength of the C–O bond decreased significantly. It is believed that the breakage of the carbamate bond resulted in the decrease in the peak strength of the C–O bond during the slow cook-off process of the HTPB film. The breakage of the C–O bond led to the formation of carbamoyl or alkoxy radicals, and the decomposition of carbamoyl radicals into ammonia radicals and CO_2_. The possible reaction process is shown in Figure 9b.

A comparative analysis of the variation of the N–H bond and C–O bond at different temperatures shows that the peak strength of the N–H bond decreased faster than that of the C–O bond, and it is believed that the breakage of the N–H bond was earlier than that of the C–O bond. The aniline structure or carbamoyl group formed by the decomposition of HTPB films was the colored group, which was the direct cause of the deepening color in HTPB film’s slow cook-off process.

### 3.3. Thermal Response of HTPE Film at Different Temperatures

The ignition temperature of the HTPE film was studied by DSC. The DSC curve showed a wide exothermic peak at 212.9 °C. The Ozawa method and Kissinger method were used to calculate the kinetic parameters of the HTPE film; they were 139.1 kJ/mol and 138.2 kJ/mol, respectively.

The surface of the HTPE film at different temperatures in the process of the slow cook-off was observed. The experimental results are shown in Figure 10. The HTPE film was light yellow and translucent at room temperature (20 °C). With the temperature increasing, the HTPE film color gradually deepened. When the temperature reached 140 °C, the HTPE film became soft and sticky, and liquid substances appeared in some areas. When the temperature reached 160 °C, the HTPE film had been completely degraded to a brown gel. In the process of the slow cook-off, the HTPE film’s binder network degraded and broke the chain through a color change analysis.

A total reflection infrared test was carried out on the HTPE film at different temperatures during the slow cook-off process (because it is liquid at 160 °C, no infrared experimental analysis was conducted on it). Infrared experimental results at room temperature are shown in Figure 11. Among them, the C–O–C stretching vibration peak was at 1276 cm^−1^, and the N–H bending vibration peak was at 1537 cm^−1^.

The change of the N–H bond with temperature is shown in Figure 12. From room temperature to 140 °C, the peak strength of the N–H bond increased firstly and then decreased gradually. When the temperature was less than 100 °C and in the slow cook-off process, the weak post-curing phenomenon of the carbamate bond happened. When the temperature was between 100 °C and 120 °C, the N–H bond was stable. When the temperature exceeded 130 °C, the N–H bond began to weaken, indicating the start of breakage.

The change rule of the C–O–C bond was studied at different temperatures in the process of the slow cook-off. The experimental results are shown in Figure 13. From room temperature to 140 °C, the C–O–C bond decreased with the increase in temperature, and this process began to weaken at 100 °C, and accelerated after 120 °C. The ether bond breakage of the HTPE film occurred after 120 °C during the slow cook-off process. The slow cook-off process of the HTPE film was a comprehensive breaking process of the carbamate bond and ether bond. The breakage of the ether bond was earlier than the carbamate bond.

### 3.4. Microstructure and Defect Statistics of Propellant during the Slow Cook-Off Process

In the process of the slow cook-off, the damage of the HTPB and HTPE propellants was visually displayed by scanning electron microscopy (SEM), and the defects of propellants were further quantitatively analyzed by the fractal dimension method [36,37]. These collected SEM images were binarized to obtain new matrix data. By using the statistical method of box-counting dimension, r squares were divided, and the number of non-empty squares N were counted. After taking the logarithms, using the correlation between logarithms, the slope was taken to obtain the box dimension, and the corresponding statistical defects were obtained by counting the box dimension values at different stages. 

The results in Figure 14 show that both the HTPB propellant binder matrix and solid filler will change with the increasing temperature. At 70 °C, the matrix structure of the HTPB propellant binder was intact, and the surface of AP particles was smooth, without the obvious interface “separation” phenomenon. At 170 °C, the N–H bond began to break, and at 190 °C, the C–O bond began to break. The main cause of increasing cracks was the breakage of carbamate bond. After 200 °C, some micro-cracks appeared in the HTPB propellant matrix. On the surface of AP particles, micro-defects appeared gradually with a large number of holes. The AP particles can be found to escape from the binder system during the slow cook-off process. Combined with the decomposition mechanism of AP mentioned above, it is considered that the defects or cracks in the HTPB propellant during the slow cook-off process were caused by three reasons. They are cracks in the binder matrix, holes in the oxidizer and “separation” at the interface.

Compared with the HTPB propellant, the HTPE propellant sample was heated to 140 °C, and a total of 6 nodes between 70 °C and 140 °C were selected. The surface defects of structural damage and box-counting dimension statistics indicated by SEM are shown in Figure 15.

Figure 15 shows that when the temperature was below 70 °C, the binder matrix structure of the HTPE propellant was complete and the surface of AP particles was smooth. There was no obvious interface “separation” phenomenon between the matrix and solid filler particles. With the temperature increasing, the most obvious cracks appeared at the interface between the binder matrix and AP particles. At 120 °C, the breakage of the C–O–C bond occurred, and the N–H bond broke at 130 °C. As a result, the HTPE binder system broke and degraded. The melt composed of the HTPE binder system’s products gradually covered the micropores of AP, and filled the defects between the AP and binder interface. When the temperature reached 140 °C, there were no obvious holes and defects (in Figure 15), and a complete and dense structure was formed again. The results of the box-counting dimension show that the defects increased firstly and decreased at 120 °C, and the maximum defect was 7.8%. When the temperature reached 140 °C, the defect reduced to only 4.2%, which is basically consistent with the surface defect of the HTPE propellant at room temperature.

### 3.5. Mechanical Properties of Propellant during the Slow Cook-Off Test

The mechanical properties of two propellants at different temperatures are shown in Figure 16, Figure 17 and Figure 18. It can be seen from Figure 16a that the maximum tensile strength of the HTPB propellant increased firstly, then decreased and increased at 190 °C. Below 150 °C, the σ_m_ increased due to the post-curing of the HTPB propellants. The post-curing increased the σ_m_ of the HTPB propellants but at a relatively slow rate. Between 150 °C and 190 °C, the maximum tensile strength began to decline, indicating that the post-curing of the HTPB propellant was complete. Combined with the FTIR experimental results, it can be seen that C–O and N–H bonds of the polyurethane in the HTPB propellant began to break at this temperature segment. The breakage resulted in a rapid decline in the maximum tensile strength of the HTPB propellant. When the temperature was higher than 190 °C, the maximum tensile strength of the HTPB propellant increased again. Oxidative crosslinking occurred as shown in Figure 16b. The vinyl in HTPB was oxidized by AP, and the double bond was opened to form peroxide bonds, indicating that oxidative crosslinking is stronger than chain-broken degradation. In the whole temperature range, the mechanical properties of the HTPB propellant are the result of the interaction between oxidation crosslinking and chain-breaking degradation.

The modulus and maximum elongation of the HTPB propellant at different temperatures are shown in Figure 17 and Figure 18. The maximum elongation decreased gradually. When the temperature was below 150 °C, the maximum elongation decreased slowly. Combined with the analysis of maximum tensile strength, it is mainly caused by post-curing. The post-curing increased the crosslinking and decreased the maximum elongation of the HTPB film. Between 150 °C and 190 °C, the maximum elongation decreased rapidly, and oxidative crosslinking played a major role. When the temperature was higher than 190 °C, the maximum elongation was stable. It indicates that the HTPB propellant had broken-chain degradation and resulted in a flat maximum elongation. The modulus was the result of the combined effect of tensile strength and elongation, which further indicates that the HTPB propellant is a combined process of oxidation crosslinking and degradation chain-breaking during slow cook-off.

Based on the above analysis, it can be seen that in the slow cook-off process of the HTPB propellant, the change in the bond network structure is influenced by two factors: oxidative crosslinking and chain-breaking degradation. When oxidative crosslinking plays a leading role, the modulus and maximum tensile strength of the HTPB propellant increase, while the maximum elongation decreases. When chain degradation plays a leading role, the modulus and maximum tensile strength decrease. In different temperature segments, the result is determined by a combination of two factors.

Compared to the experimental results of the HTPB propellant, the mechanical properties of the HTPE propellant can be seen in Figure 16, Figure 17 and Figure 18. Under different temperatures, the maximum tensile strength and maximum elongation of the HTPE propellant decreased gradually. Compared with the HTPB propellant, HTPE propellant also had a post-curing phenomenon, but the post-curing phenomenon was very weak. Combined with those FTIR experimental results, it can be seen that the C–O–C ether bond of the HTPE propellant was broken and then melts were formed. The breakage of bonds and melts formed by products result in a significant decrease in the maximum tensile strength and maximum elongation. However, the modulus of the HTPE propellant increases firstly and then decreases. It is the result of the interaction of the C–O–C ether bond and N–H bond.

### 3.6. Change of Propellant Gel Fraction

The gel fraction of a solid propellant refers to the mass ratio between the gel generated after solidification of the binder system and the total binder system in the solid propellant. This parameter is related to the properties of the propellant elastomer network under certain conditions, so the measurement of this parameter has practical significance for the study of the mechanical properties and storage aging properties of a propellant [38,39,40]. The changes in gel fraction of the HTPB and HTPE propellants during the slow cook-off are compared in Figure 19.

It can be seen that the gel fraction of the HTPB propellant increased firstly, and decreased at 190 °C. When the temperature got to 200 °C, the number of gel fractions increased around 80%. The main factors affecting the change of propellant gel fraction are post-curing, oxidative crosslinking and chain-breaking degradation of the binder network. Post-curing and oxidative crosslinking will increase the gel percentage, while the degradation of chain breaking will decrease the gel percentage. In the slow cook-off process of the HTPE propellant, the gel fraction decreased with the increasing temperature due to the breakage of the ether bond.

### 3.7. Change Analysis of Propellant Slow Cook-Off Process

During the slow cook-off test, the cracks in the binder matrix, the holes in the oxidizer and the “separation” of the interface are the three reasons for defects or cracks in the HTPB propellant. In the heating process, increasing defects made the burning surface of the HTPB propellant increase sharply. Figure 20 shows the schematic diagram of defect and crack formation. The breakage of the HTPB binder occurred at 170~190 °C. When the temperature rose to about 200 °C, AP holes appeared, and small white spots can be seen on gray balls in the diagram. When the temperature rose further to 230 °C, further breakage of the HTPB binder occurred, and AP separated from the fractured crosslinked network, resulting in an explosion of the HTPB propellant.

After the HTPE propellant was heated to 120 °C, the products generated by the breakage of the carbamate bond and ether bond formed a “high temperature self-repair body”. The “high temperature self-repair body” covers the surface of AP and will repair the defects and holes of AP and interface. The holes and cracks of the HTPE propellant can also be filled by these melts. The burning surface of the HTPE propellant will not increase greatly. The “high temperature self-repair body” covering on AP reduces the sensitivity of AP to pressure, so that the burning rate and burning surface of the HTPE propellant will not increase sharply at a high temperature. The internal pressure of the solid engine will not rise sharply because of the “high temperature self-repair body”. Thus, the response intensity of the HTPE solid engine in the slow cook-off test was reduced and its low vulnerability was improved.

## 4. Conclusions

(1) The ignition of the HTPB propellant in the process of the slow cook-off is mainly caused by the hole of AP at 200 °C and the separation of AP from the binder system. The ignition temperature of the HTPE propellant is mainly affected by the HTPE binder. In the slow cook-off process, the carbon–carbon double bond on the HTPB polymer chain is opened and oxidized crosslinking occurs. With the occurrence of post-curing, the C–O bond and N–H bond have a tendency of increasing firstly and then decreasing and breaking at 170~190 °C, which separates the solid particles originally attached to the crosslinking network. The post-curing phenomenon of the HTPE propellant is weak, and the decomposition of the C–O–C ether bond and N–H bond happens during the slow cook-off. These products will melt after 140 °C, and coat and fill between solid particles.

(2) The mechanical properties of the solid propellant change with the slow cook-off temperature: under the combined effect of oxidative crosslinking and degradation chain-breaking, the modulus of the HTPB propellant increases gradually; the maximum elongation decreases gradually with the increasing temperature; and the maximum tensile strength increases firstly, decreases secondly and then increases. The HTPE propellant mainly undergoes bond breakage and decomposition, which makes the modulus and maximum tensile strength decrease gradually, and the maximum elongation increase continuously.

(3) In the slow cook-off process, the gel fraction of the HTPB propellant increases due to post-curing and oxidative crosslinking and decreases due to chain breakage. Compared with the HTPB propellant, the gel fraction of the HTPE propellant decreases continuously due to the breakage of the ether bond. In the microscopic analysis, the gap and crack of the HTPB propellant always increase, while the HTPE propellant shows the effect of melt filling and decreases at the end. The analysis concludes that the breakage of urethane and ether bonds of the HTPE binder system occurs after 120 °C, which is the root cause of the difference in thermal response of the two propellants. The HTPE propellant will then generate a “high temperature self-repair body”; the molten products cover the defects and holes of the AP and interface. Because of that, the degree of response of the HTPE propellant will be reduced, and the low vulnerability of the HTPE propellant is improved.

## Figures and Tables

**Figure 1 polymers-14-03699-f001:**
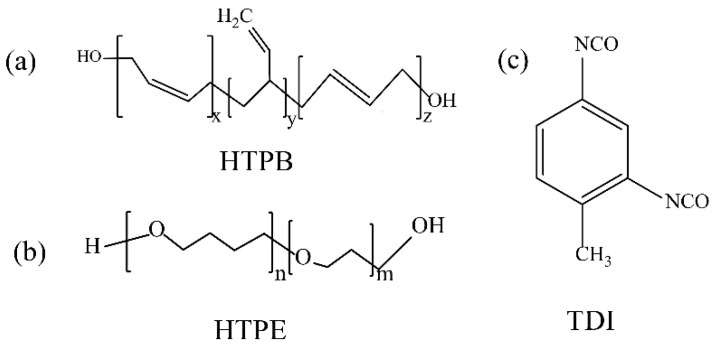
Structural formulae of HTPB, HTPE and TDI.

**Figure 2 polymers-14-03699-f002:**
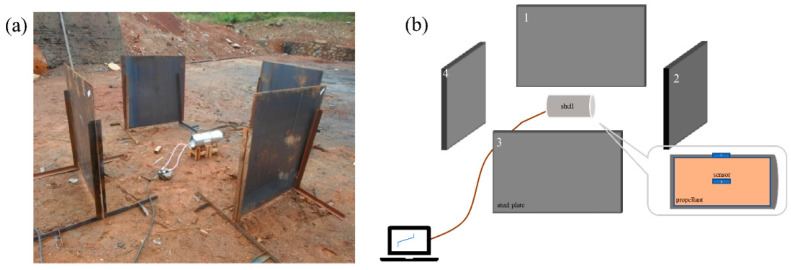
(**a**) The scaled-down solid engine slow cook-off device diagram and (**b**) schematic diagram.

**Figure 3 polymers-14-03699-f003:**
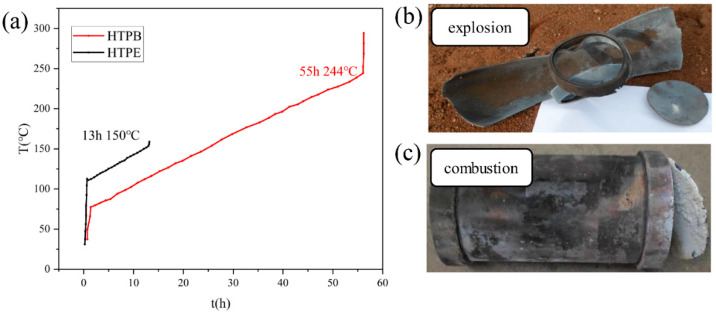
(**a**) The scaled-down solid engine slow cook-off test curve, (**b**) the wreckage of HTPB propellant, and (**c**) the wreckage of HTPE propellant.

**Figure 4 polymers-14-03699-f004:**
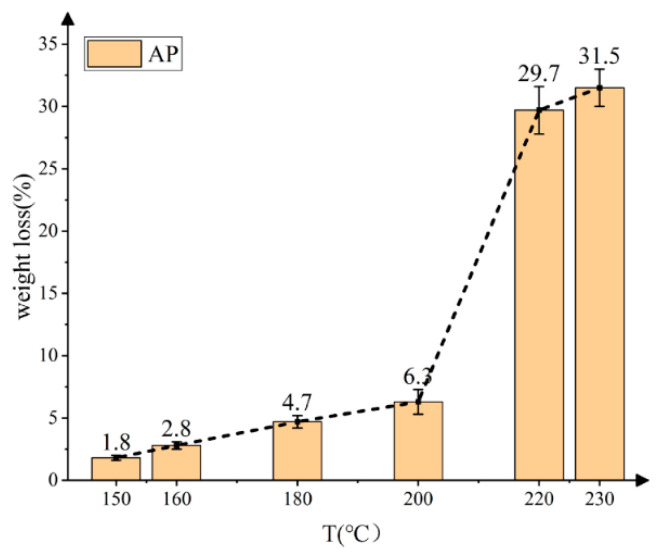
The mass-loss rate of AP with temperature.

**Figure 5 polymers-14-03699-f005:**
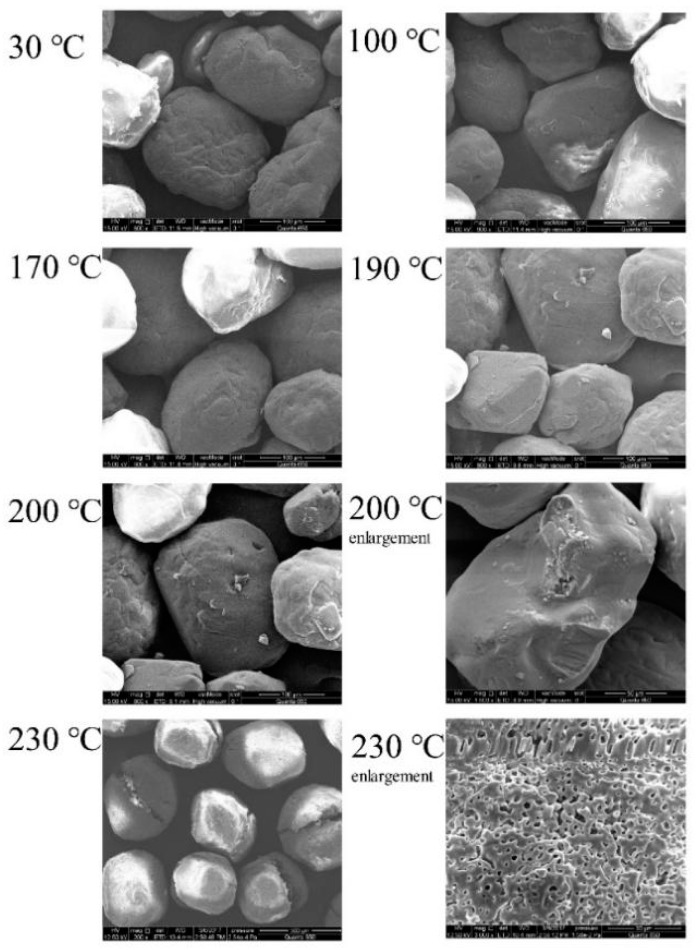
AP surface conditions at different temperatures.

**Figure 6 polymers-14-03699-f006:**
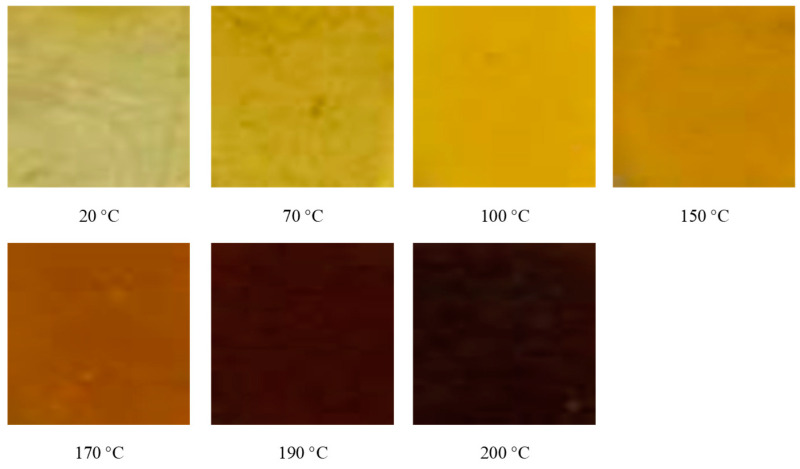
The morphology of HTPB film at different temperatures.

**Figure 7 polymers-14-03699-f007:**
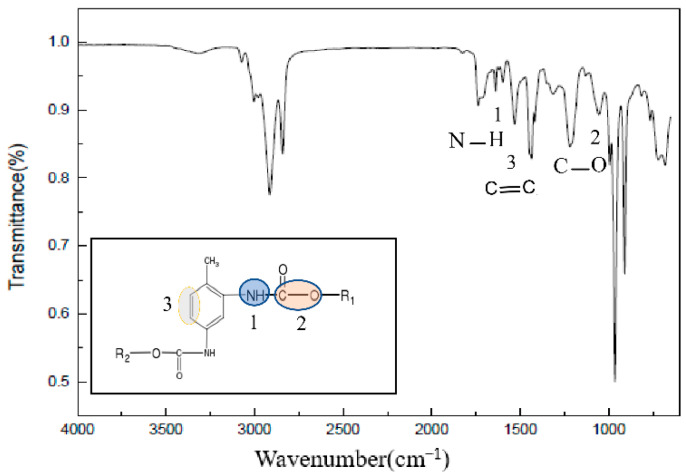
The infrared spectrum of HTPB film at room temperature.

**Figure 8 polymers-14-03699-f008:**
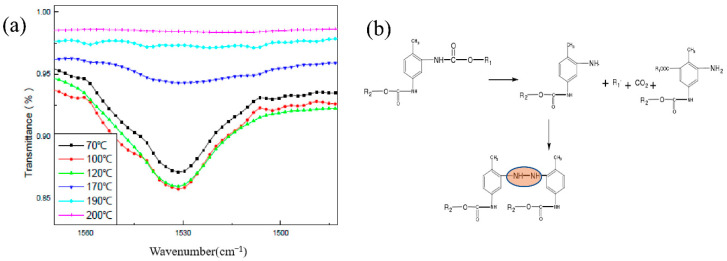
(**a**) Variation trend of N–H relative absorbance and (**b**) decomposition reaction process.

**Figure 9 polymers-14-03699-f009:**
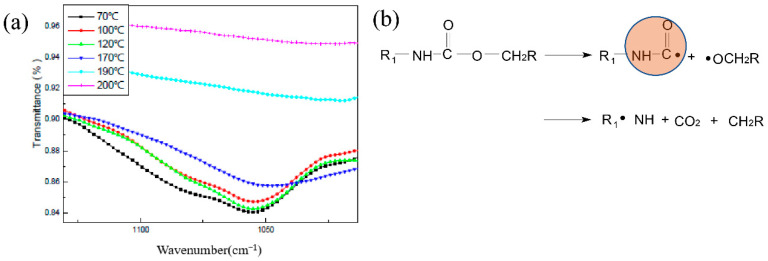
(**a**) Variation trend of C–O relative absorbance and (**b**) decomposition reaction process.

**Figure 10 polymers-14-03699-f010:**
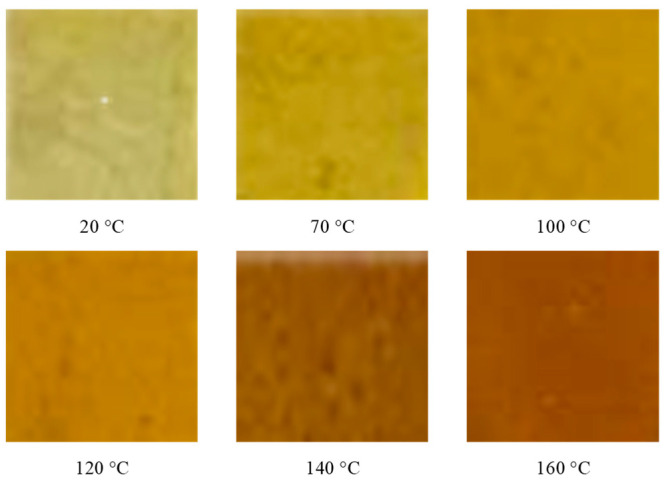
The morphology of HTPE film at different temperatures.

**Figure 11 polymers-14-03699-f011:**
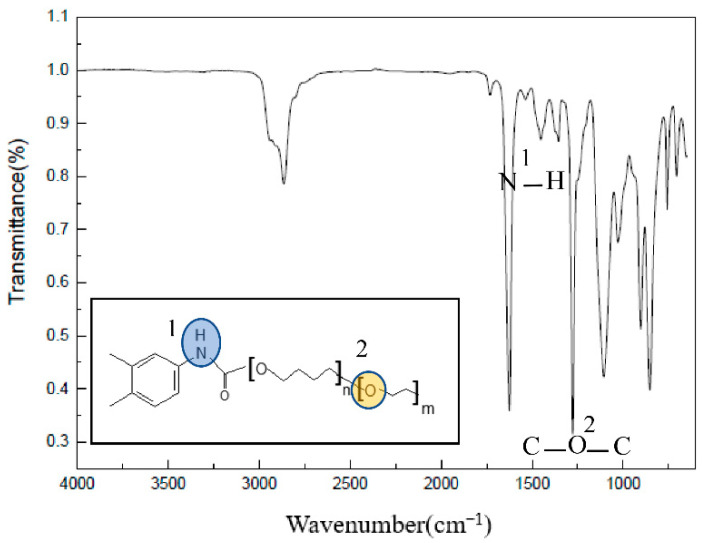
The infrared spectrum of HTPE film at room temperature.

**Figure 12 polymers-14-03699-f012:**
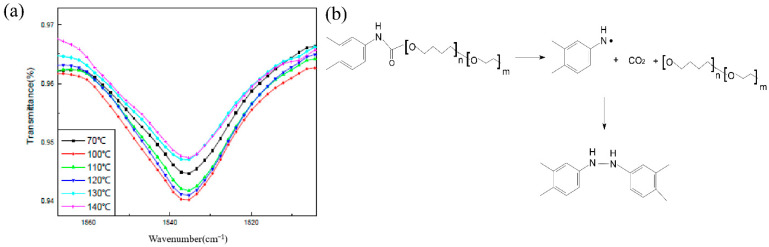
(**a**) Change trend of N–H relative absorbance and (**b**) decomposition reaction process.

**Figure 13 polymers-14-03699-f013:**
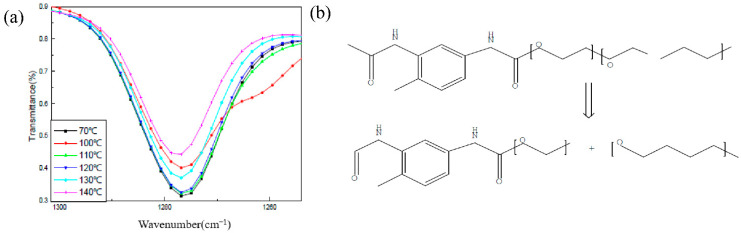
(**a**) Change trend of C–O–C relative absorbance and (**b**) decomposition reaction process.

**Figure 14 polymers-14-03699-f014:**
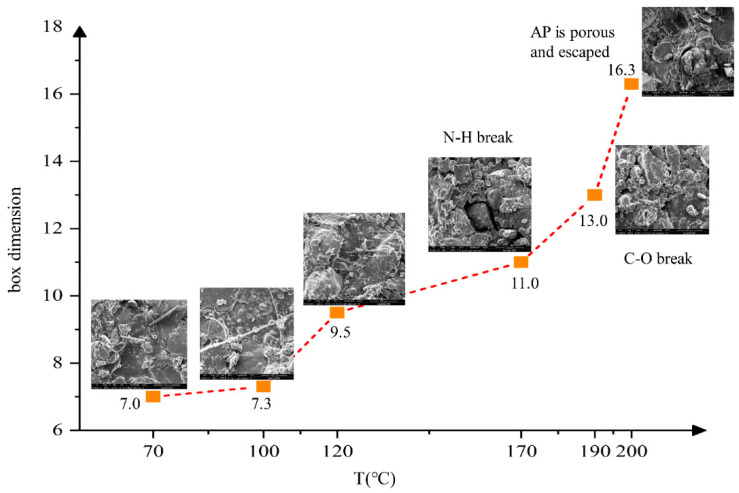
Defect figure of HTPB propellant at different temperatures during slow cook-off.

**Figure 15 polymers-14-03699-f015:**
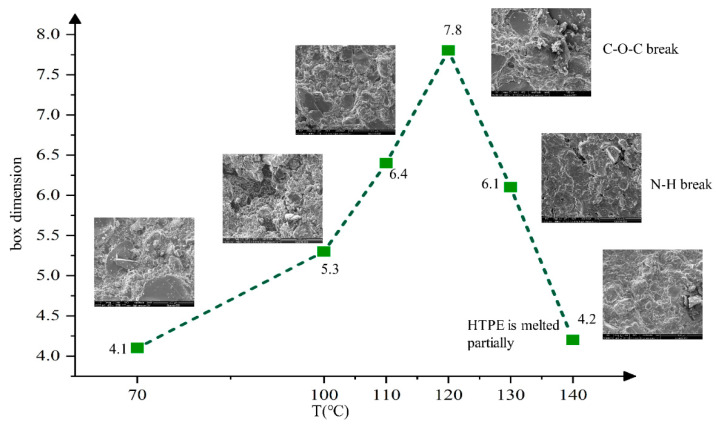
Defect figure of HTPE propellant at different temperatures during slow cook-off.

**Figure 16 polymers-14-03699-f016:**
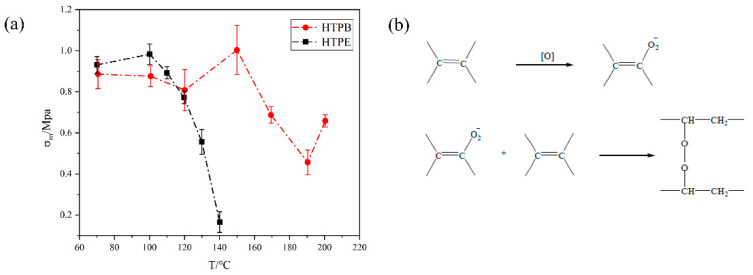
(**a**) The relationship between maximum tensile strength and temperature in HTPB and HTPE propellant (**b**) oxidation of C=C in HTPB.

**Figure 17 polymers-14-03699-f017:**
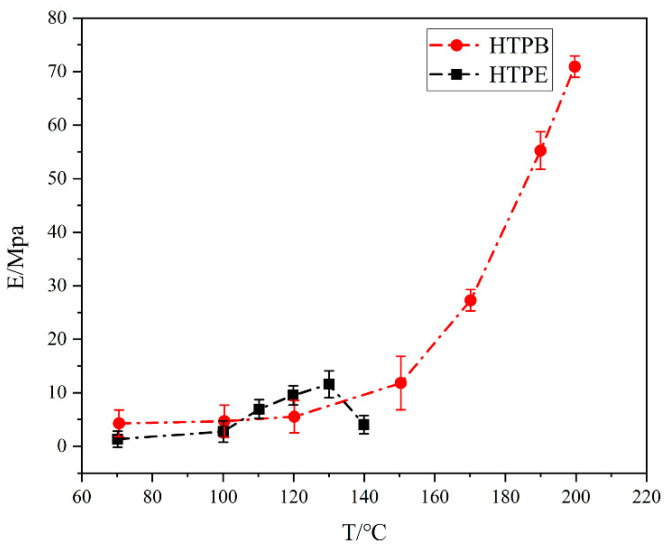
The relationship between modulus and temperature in HTPB and HTPE propellant.

**Figure 18 polymers-14-03699-f018:**
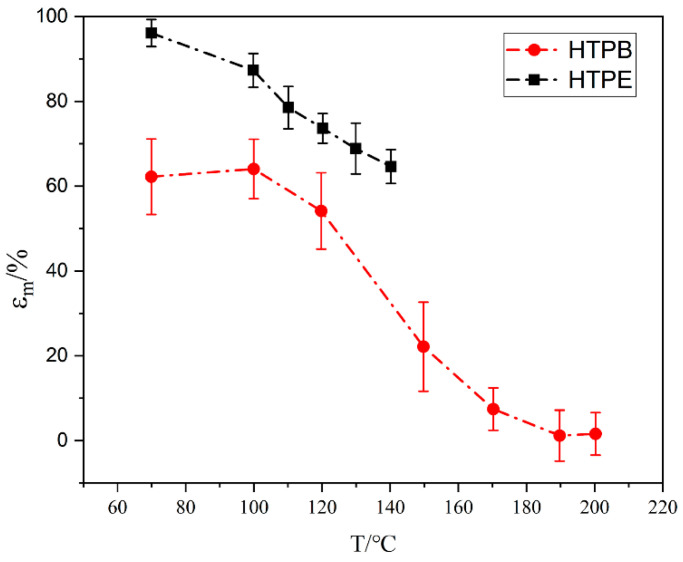
The relationship between maximum elongation and temperature in HTPB and HTPE propellant.

**Figure 19 polymers-14-03699-f019:**
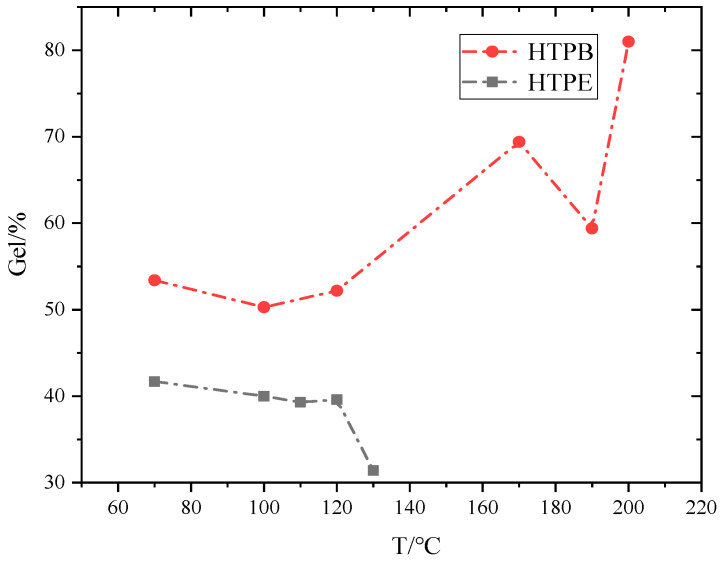
The relationship between gel fraction and temperature of HTPB and HTPE propellant.

**Figure 20 polymers-14-03699-f020:**
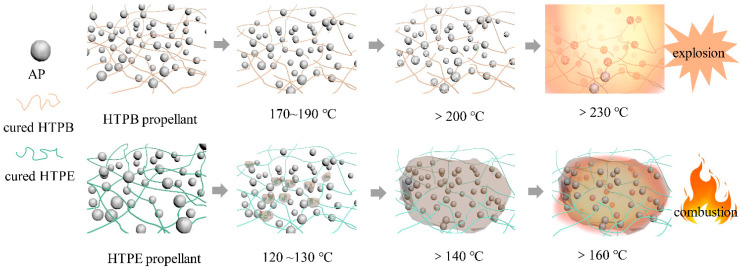
The evolution of propellant defects and repair.

## Data Availability

The data presented in this study are available on request from the corresponding author.

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
