# Peer review of "Comparative Study on Thermal Response Mechanism of Two Binders during Slow Cook-Off"

_polymers, 2022, doi:10.3390/polym14173699_

Round 1
Reviewer 1 Report
Dear Authors,
After reading the article, I would like to congratulate the authors on the accuracy of its implementation and the very thoughtful discussion part.
I'm just wondering if you could give the elemental or oxide composition in a quantitative context for selected materials (HTPE & HTPB) to specify the potential standard deviations from the obtained results. However, this is a remark that could be included in the next publication, and I think they will be. I believe that the article was prepared with great commitment.
For your message I'm sending 1 more article which may be useful in your reflections.
https://www.mdpi.com/2073-4360/11/4/598/htm
Thank You,
Reviewer
Author Response
Dear Reviewer:
I am very grateful to your comments for the manuscript. It is a good idea to study about the elemental or oxide composition in a quantitative context for selected materials (HTPE & HTPB) to specify the potential standard deviations from the obtained results. I will continue my research in this direction, and may you will find them in my next manuscript. According with your advice, this article, https://www.mdpi.com/2073-4360/11/4/598/htm, was read carefully, it is very useful, and cited in my new manuscript.

Reviewer 2 Report
The subject of the article Comparative study on thermal response mechanism of two binders during slow cook-off raises important issues both from an application and a scientific point of view.
The article is well structured. The following are some minor notes:
Abstract
In my opinion, it is too extensive, it gives very detailed results. I suggest rebuilding.
Introduction
This section, on the other hand, is quite short (18 references are cited in this section), but the description of the issues undertaken by the authors is sufficient. At the end of this part, the Authors justified the purpose of the research, stated that in the research area described in the Introduction, there are no detailed analyzes of this type of research.
Materials and methods
This part is correct. However, there is no information on the number of samples used in each study.
Results and Discussion
The research was well described and the results thoroughly discussed and documented in the form of photos and drawings. The research methods do not raise any objections, but there is a lack of a statistical approach to the obtained results - dispersion measures (e.g. standard deviation) - error bars appeared only in Figure 4. In Figs. 17 and 18 - no description of the vertical axis.
Conclusions
The conclusions are presented in a concise manner and adequate to the obtained results.
References
The article contains 32 references, most of them are up-to-date. No reference to item 20 in literature was found in the text of the article - the article is thematically very similar to the reviewed one.
Author Response
Dear reviewer:
I am very grateful to your comments for the manuscript. These comments are very useful and a point-by-point response have been written. Please see the attachment.

Reviewer 3 Report
The present manuscript is very intresting.
A minor flaw regards the literature review, which should be strengthened in order to link the proposed study with the aims and scope and the audience of Polymers, by more articles.
In the abstract, the present reviewer proposes to introdyce the meanings of HTPB and HTPE.
In the line 76, after HTPB brinder there is a notation (III). similar in line 81 for AP. Could you please explain the meaning and the reason?
In the part 2.2, in the lines 101-106, there is a presentation of the samples. The authors compared the behavior of the HTPB and HTPE but they do not have the same % in the samples. For HTPB there is binder/AP/Al/auxiliary = 10/70/12/2 but for HTPE binder/AP/Al/auxiliary= 20/72/5/3. The reviewer would like to know why you did not have for example Al or auxiliary constant.
Why experiments in outer surface? Is it possible to have the temperature stable and homogeneous?
Figure 13b needs to improved in terms of quality.
In figure 20, for HTPB it is not very clear the differences in the figure of 170-190oC and >200oC.
The supplement information pages 19-20, gives some diagrames that are confusing? Green line and blue line present...? As there is no explanations, the reviewer proposed to delete these pages or include them in the main part of the manuscript.
Author Response
Dear reviewer:
I am very grateful to your comments for the manuscript. These comments are very correct and useful, a point-by-point response to these comments have been written in the attachment. Please see the attachment, and thanks for these helpful comments again.

Round 2
Reviewer 3 Report
The present reviewer propose the qcceptqnce of the revised manuscript in present form.